# Towards Efficient Automatic Self-Pruning of Large Language Models

## Abstract

Despite exceptional capabilities, Large Language Models (LLMs) still face deployment challenges due to their enormous size. Post-training structured pruning is a promising solution that prunes LLMs without the need for retraining, reducing computational overhead, and it is hardware-deployment friendly. However, the training-free nature of post-training structured pruning leads to significant performance degradation. We argue that the key to mitigating this issue lies in accurately determining the pruning rate for each layer. Meanwhile, we find that LLMs may have prior knowledge about their own redundancy. Based on this insight, we introduce **Self-Pruner** an end-to-end automatic self-pruning framework for LLMs, which efficiently search layer-wise pruning rates. Specifically, **Self-Pruner** leverages LLMs to autonomously execute the entire evolutionary search process to search for pruning rate configurations. In this process, LLMs are used to generate populations, select parent solutions from the current population, and perform crossover and mutation operations to produce offspring solutions. In this way, LLMs automatically generate and evaluate a large number of candidate solutions, effectively converging to find the pruning rate configurations with minimal human intervention. Extensive experiments demonstrate **Self-Pruner**'s better performance compared to existing state-of-the-art methods. Notably, **Self-Pruner** prunes LLaMA-2-70B to 49B level with only 0.80% drop in accuracy across seven commonsense reasoning tasks, achieving a $1.39\times$ speedup on NVIDIA A100 80GB GPU. Further pruning to 35B level resulted in only a 3.80% decrease in accuracy while obtaining a $1.70\times$ speedup. Code is available in the supplementary material.

## 1 Introduction

In recent years, with the rapid advancement of Large Language Models (LLMs) (Zhang et al., 2022; Touvron et al., 2023a;b), these models have achieved remarkable performance in language understanding and generation (Brown et al., 2020; Wei et al., 2022; Lewis et al., 2020). However, the dramatic increase in the number of parameters has led to significant rises in computational resource consumption and deployment costs (Zhu et al., 2023). To maintain model performance while mitigating computational complexity, various model compression techniques such as pruning (Frantar & Alistarh, 2023; Sun et al., 2023; Ashkboos et al., 2024; Dong et al., 2024), quantization (Egiazarian et al., 2024; Xiao et al., 2023; Huang et al., 2024; Shao et al., 2024), and knowledge distillation (Agarwal et al., 2024; Gu et al., 2024; Ko et al., 2024; Wan et al., 2024) have emerged. Among these, structured pruning (Ma et al., 2023; Li et al., 2024; An et al., 2024) stands out as it not only significantly reduces computational overhead and memory usage but also enhances inference speed on various hardware platforms. This dual benefit of efficiency and accelerated inference has paved the way for more widespread practical applications of LLMs (Ma et al., 2023; Muralidharan et al., 2024b).

Traditional structured pruning methods often involve retraining the model, including but not limited to training from random initialization (Wang et al., 2020), fine-tuning the pruned model (Hou et al., 2020), or performing iterative pruning (Zhu & Gupta, 2017; Molchanov et al., 2019). However, the inherent complexity of LLMs and their substantial demands for computational resources and data make these traditional retraining-required structured pruning strategies difficult to implement in practice (Xia et al., 2023; Muralidharan et al., 2024a). As a result, post-training pruning has

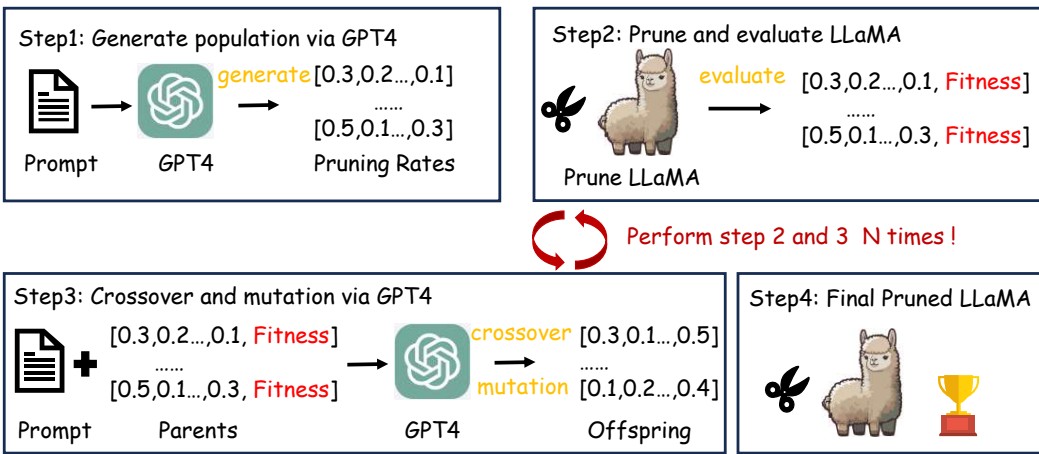

Figure 1: An overview of Self-Pruner. Self-Pruner first instructs LLMs with prompts to generate layer-wise pruning rates as the initial population. Next, Self-Pruner uses these layer-wise pruning rates to prune the model, evaluating the pruned model to obtain each individual's fitness. Then, Self-Pruner instructs LLMs to select parent individuals for crossover and mutation, generating offspring. This process is repeated for N evolutionary iterations to obtain the final pruned model.

emerged as an increasingly important alternative. This approach is particularly advantageous when pruning LLMs due to its minimal resource requirements (Zhang et al., 2023b; Dong et al., 2024).

However, it is precisely due to the training-free nature of post-training pruning that leads to a severe decrease in the accuracy, especially in structured pruning. We observe that this is largely caused by the inaccurate setting of layer-wise pruning rate. A straightforward way to set the layer-wise pruning rate is to apply a uniform pruning rate to each layer (Sun et al., 2023; Frantar & Alistarh, 2023). For instance, LLM-Pruner (Ma et al., 2023) applies the same pruning rate to all the pruned layers. However, this approach is suboptimal, as the contribution of each layer to the final accuracy varies significantly, applying a uniform pruning rate across all layers risks removing important weights (Cheng et al., 2024; Yang et al., 2023). OWL (Yin et al., 2023) has already recognized this issue and adopted a heuristic metric to set the pruning rate of LLMs inversely proportional to the observed ratio of abnormal activations within each layer, thereby achieving non-uniform pruning of LLMs. However, the OWL method relies on manually designed importance metrics, requiring human expert involvement and tedious iterative experimentation. Additionally, it demands meticulous tuning of hyperparameters to achieve best performance, making it inefficient.

In this paper, we propose an end-to-end automatic LLMs pruning framework named Self-Pruner, which efficiently searches for layer-wise pruning rates, significantly enhancing the quality of post-training pruning for LLMs. Self-Pruner is a framework that uses evolutionary algorithm (Holland, 1992; Bäck et al., 1997) to search for layer-wise pruning rates. Evolutionary algorithms have been used to automate pruning of CNNs (Liu et al., 2019; Salehinejad & Valaee, 2021) and Transformers (Li et al., 2022; Liu et al., 2024b). However, the above algorithm requires numerous iterations to converge to the final solution, which is impractical for LLMs with billions of parameters, as evaluating the performance of pruned LLMs is highly time-consuming (Chang et al., 2024). Meanwhile, the success of evolutionary algorithms for automate pruning depends largely on the design of the algorithm. For different pruning networks and compression constraints, specialized genetic operators (such as crossover and mutation) need to be customized (Liu et al., 2019; Shang et al., 2022). Designing these evolutionary algorithms manually is often time-consuming and requires extensive experience and knowledge (Liu et al., 2024c; Lange et al., 2024).

To accelerate the convergence of the evolutionary search and achieve automation in algorithm design, we chose to have the LLM execute the entire evolutionary algorithm process itself. The insight behind this approach is that we found LLMs may possess prior knowledge about their own redundancy (Dong et al., 2022; Zhang et al., 2023a; Zheng et al., 2023). We can take advantage of this by having LLMs generate and evaluate feasible solutions. Therefore, Self-Pruner uses LLMs to gen-

erate the initial population, constructs a prompt to guide LLMs to select parent solutions from the current population and perform crossover and mutation to generate offspring solutions. These new solutions are then evaluated and added to the population for the next round by LLMs. In this way, LLMs can automatically generate and evaluate a large number of candidate solutions, quickly converging to find the superior pruning rate configurations with minimal human intervention. Notably, we achieved self-pruning of LLMs through evolutionary search process, marking an important step towards the fully automated compression of LLMs.

Extensive experiments on language modeling and zero-shot tasks demonstrate that Self-Pruner achieves better performance compared to existing post-training pruning methods. Using the Self-Pruner method to prune the LLaMA-2-70B (Touvron et al., 2023b) model, we obtained a 49B model that achieved only 0.80% drop in average zero-shot accuracy across seven commonsense reasoning tasks as LLaMA-2-70B with a 1.39× increase in inference speed on GPU. Further pruning resulted in a 35B model shows only a 3.80% decrease in accuracy compared to LLaMA-2-70B while achieving a 1.70× increase in inference speed. Both results represent state-of-the-art performance in existing post-training structured LLMs pruning. The main contributions of this paper are summarized as follows:

- We propose a novel end-to-end automatic pruning framework that utilizes LLMs to efficiently search for layer-wise pruning rate without human intervention, marking significant step toward fully automated LLMs compression.
- We propose a novel method which leverages LLMs to execute the entire evolutionary search process, including population generation, selection, crossover, and mutation, enabling the self-pruning of LLMs.
- Extensive experiments show that Self-Pruner outperforms existing post-training pruning methods, achieving competitive accuracy, while significantly reducing model size.

## 2 RELATED WORK

**Post-training Structured Pruning of LLMs.** Structured pruning (Ma et al., 2023) offers advantages such as hardware-friendly sparsity patterns and reduced memory footprint. Traditionally, structured pruning methods required retraining the model, which was effective for smaller networks (Molchanov et al., 2016; Hou et al., 2020). However, when it comes to LLMs, retraining becomes impractical due to the enormous computational resources and time required (Xia et al., 2023; Muralidharan et al., 2024a). This challenge has led to the development of post-training pruning techniques specifically tailored for LLMs. Post-training pruning aims to reduce model size and inference time without the need for extensive retraining (Ma et al., 2023; Sun et al., 2023). However, existing post-training structured pruning techniques can lead to a sharp drop in the accuracy of LLMs (Ma et al., 2023; An et al., 2024). In this work, we found that the layer-wise pruning rate configuration has a significant impact on the accuracy of post-training structured pruning for LLMs. Through carefully searched layer-wise pruning rates, we can greatly improve the accuracy of existing post-training structured pruning LLMs.

**LLMs for Optimization.** In recent years, the capabilities of LLMs have significantly improved (Naveed et al., 2023). People can now use LLMs to help solve a wide range of problems, including optimization tasks. For example, LLMs have been employed in heuristic algorithm design (Liu et al., 2024a; Romera-Paredes et al., 2024), prompt optimization (Yang et al., 2024), solving black-box optimization problems (Liu et al., 2024d; Song et al., 2024), and neural architecture search (Zheng et al., 2023; Chen et al., 2024). LLMs have demonstrated powerful understanding and reasoning capabilities (Brown et al., 2020; Wei et al., 2022) in the aforementioned optimization domains. Naturally, this leads us to consider whether LLMs can be used to optimize the model pruning problem, specifically by having LLMs design an excellent pruning model. Due to the extensive training of LLMs on massive amounts of data, they encompass a wide range of domain knowledge, enabling them to integrate knowledge from multiple related fields (Madani et al., 2023; Hong et al., 2023) and thereby propose more comprehensive and effective pruning strategies.

**LLMs meet Evolutionary Algorithms.** Evolutionary algorithms are a class of optimization algorithms that simulate biological evolutionary processes, solving complex optimization problems

by mimicking mechanisms such as natural selection, inheritance, and mutation (Eiben et al., 2015; Bartz-Beielstein et al., 2014). The integration of evolutionary computation into prompt engineering for LLMs has shown great potential in improving performance across multiple domains. For example, it has been applied to code generation (Liventsev et al., 2023; Lehman et al., 2023), text generation (Guo et al., 2023; Xu et al., 2023), and heuristic algorithm design (Liu et al., 2024a; Romera-Paredes et al., 2024). The success in these areas have inspired our idea to apply the combination of evolutionary algorithms and LLMs to automated model pruning. By allowing LLMs to perform the evolutionary search process, we anticipate gradually optimizing pruning strategies through iterations, thereby effectively enhancing the accuracy of existing pruning methods. This approach not only fully utilizes the powerful reasoning capabilities of LLMs (Huang & Chang, 2022) but also leverages the advantage of evolutionary algorithms in finding solution within complex search spaces (Whitley, 2001), providing a novel perspective for addressing the challenging problem of model pruning.

## 3 METHODOLOGY

### 3.1 PRELIMINARIES

In post-training pruning of LLMs, a certain proportion of pre-trained weights needs to be removed to obtain the pruned LLMs. Since the redundancy varies across different layers of the LLMs, the number of parameters removed from each layer has a significant impact on the accuracy of the pruned LLMs (Yin et al., 2023; Xu et al., 2024). Given a pre-trained LLM with $n$ layers, we define the layer-wise pruning rates as $\boldsymbol{p} = (p_1, p_2, ..., p_n)$, where $0 \leq p_i \leq 1$ represents the pruning rate of the $i$-th layer, i.e., the ratio of the remaining parameters after pruning to the original number of parameters in that layer. Our goal is to find the layer-wise pruning rates $\boldsymbol{p}^*$ such that the pruned LLM achieves the best accuracy on the test set, while ensuring that satisfies the constraint on the average pruning rate, i.e., $\frac{1}{n} \sum_{i=1}^{n} p_i = \beta$, where $\beta$ is the given average pruning rate. This problem can be formulated as an optimization problem as bellow:

$$(p_1, p_2, ...p_n)^* = \underset{p_1, p_2, ...p_n}{\arg\max} \; \mathrm{acc}(\mathrm{LLM}(p_1, p_2, ...p_n))$$
$$s.t. \quad \frac{1}{n} \sum_{i=1}^{n} p_i = \beta, \tag{1}$$

where $\mathrm{LLM}(\cdot)$ denotes the pruned LLM with layer-wise pruning rate $(p_1, p_2, ...p_L)$, and $\mathrm{acc}(\cdot)$ refers to the accuracy of the pruned LLM on the test set. Since this optimization problem is generally intractable, we employ evolutionary algorithm to search for $\boldsymbol{p}^*$. Although evolutionary algorithm have been successfully applied to optimize pruning rates in CNNs (Liu et al., 2019; Lin et al., 2020; Shang et al., 2022) or Transformers (Li et al., 2022; Liu et al., 2024b), we improve the search process of the evolutionary algorithm by leveraging LLMs.

### 3.2 SELF-PRUNER ALGORITHM

The Self-Pruner algorithm framework is illustrated in Figure 1 and detailed in Algorithm 1. Overall, Self-Pruner uses LLMs to generate the initial population and employs model perplexity as the fitness metric for individuals. LLMs then select parents for mutation and crossover to generate offspring. This process is repeated for N iterations to obtain the final pruning rate configuration and model fitness. The Self-Pruner algorithm consists of several stages, which we describe below:

**Population initialization.** Self-Pruner utilizes LLMs to generate the initial population for evolutionary search, a process assisted by carefully constructed prompts. The prompt is shown in Figure 2. Specifically, the prompt consists of two parts:

- **Problem description and task instruction:** this part describes the problem that LLMs are instructed to solve, namely to assist in model pruning by outputting layer-wise pruning rate configurations.
- **Solution attributes:** this section specifies some fundamental attributes that the new solutions generated by LLMs should adhere to.

---

### Problem description and task instruction ###
Let's think step by step! You are helping me prune the {model}, aiming to minimize perplexity on the WikiText-2 dataset. The model has {number of model layers} transformer layers. Layer-wise pruning rate measures how many parameters are pruned from each layer of the model. Different layers may have different pruning rates based on their importance and contribution to the performance of model. You need to generate {population size} valid layer-wise pruning rate configurations. Each configuration should:
### Solution attributes ###
· Contain {number of model layers} decimals between 0 and 1, accurate to 5 decimal places.
· Ensure the average of these numbers equals {pruning ratio}.
· Be distinct, starting with "[" and ending with "]".
Your response should only contain the {population size} configurations without any additional text.
Note: {} is used to indicate placeholders.

---

Figure 2: Prompt for population initialization.

Leveraging the prior knowledge that LLMs inherently possess about model architecture can generate a high-quality initial population. Compared to random initialization, this method accelerates the search by providing high-quality initial solutions.

**Select Individuals based on Fitness.** After generating the layer-wise pruning rate configuration, we determine the number of parameters to prune for each layer based on the layer-wise pruning rate and use the Wanda-sp (An et al., 2024) metric to identify which neurons within the layer should be pruned. Additionally, we evaluate the pruned LLM on the WikiText-2 (Merity et al., 2016) dataset to obtain the perplexity of the pruned LLM, which serves as the fitness metric. In this context, a lower perplexity score indicates higher fitness. We select individuals based on their fitness.

---

### Problem description and task instruction ###
Let's think step by step! You will receive {population size} lists representing the layer-wise pruning rates of the {model} and a fitness value for each list. The lower the fitness value, the better. Your task is to perform the mutation/crossover operation in the evolutionary algorithm to generate new configurations. Each new pruning rate configuration list should:
### Solution attributes ###
· Contain {number of model layers} decimals between 0 and 1, accurate to 5 decimal places.
· Ensure the average of these numbers equals {pruning ratio}.
· Be distinct, starting with "[" and ending with "]".
Please provide exactly {number of mutation/crossover} new configurations based on the existing data provided below without any additional text.
### Current Population and Fitnesses ###
Here are the existing layer-wise pruning rate configurations and their fitness values:
Configuration1: {layer-wise pruning rate}, Fitness1: {fitness}
Configuration2: {layer-wise pruning rate}, Fitness2: {fitness}
...
Note: {} is used to indicate placeholders.

---

Figure 3: Prompt for crossover and mutation.

**Mutation and Crossover.** Self-Pruner leverages LLMs to execute key steps of the evolutionary search: parent selection, crossover, and mutation. This process is guided by a carefully crafted prompt, as illustrated in Figure 3. The prompt consists of three critical components:

- **Problem Description and Task Instructions:** this part instructs the LLMs to perform parent selection, crossover, and mutation operations to generate new offspring.
- **Solution attributes:** provides detailed guidelines on the attributes and format requirements that LLMs must adhere to when generating new solutions. This part is consistent with the prompt used in population initialization.

- **Current Population and Fitnesses:** provide information about the individuals in the current population and their corresponding fitness values, allowing LLMs to select individuals for crossover and mutation.

The uniqueness of Self-Pruner lies in its approach of not guiding the LLM through detailed algorithmic steps for precise mutation and crossover operation, but instead using high-level natural language instructions. This approach significantly reduces the human effort and tedious trial-and-error required for designing mutation and crossover operators, enhancing the method's generality and flexibility. Based on its understanding of the problem and the provided context, LLMs autonomously perform selection, crossover, and mutation operations to generate new, potentially pruning configuration schemes.

We present the specific algorithmic process of Self-Pruner in Algorithm 1. Self-Pruner begins by initializing a population $\mathcal{G}_0$ of $\mathcal{K}$ layer-wise pruning rate configurations using LLMs and the individuals in the population all satisfy the constraint which the average pruning rate is equal to $\beta$ (Line 1). It then proceeds through $\mathcal{N}$ evolutionary iterations (Line 2-8). In each iteration, the algorithm evaluates the fitness of the individuals (Line 3) and selects the top $\mathcal{K}$ individuals (Line 4). Self-Pruner utilizes LLMs to select individuals for mutation (Line 5) and crossover (Line 6) to generate offspring. These offspring are added to the population for the next selection (Line 7). This iterative process continues until the predefined number of iterations is reached. Ultimately, Self-Pruner outputs the final layer-wise pruning rate configuration (Line 9-10).

---

**Algorithm 1** Self-Pruner

---

**Hyper Parameters**: Population Size: $\mathcal{K}$, Number of Mutation: $\mathcal{M}$, Number of Crossover: $\mathcal{S}$, Max Number of Iterations: $\mathcal{N}$ .
**Input**: Pre-trained LLM: LLM, Average pruning rate: $\beta$ .
**Output**: The best found pruning rates: $\boldsymbol{p}^*$ with fitness Fitness$^*$.

1: $\mathcal{G}_0$ = Initialization($\mathcal{K}$), s.t. $\beta$;
2: **for** $i = 0 : \mathcal{N}$ **do**
3:     $\{\mathcal{G}_i, \text{Fitness}\}$ = Inference(LLM($\mathcal{G}_i$));
4:     $\mathcal{G}_i$ = Top $\mathcal{K}(\{\mathcal{G}_i, \text{Fitness}\})$;
5:     $\mathcal{G}_{mutation}$ = Mutation($\mathcal{G}_i, \mathcal{M}$), s.t. $\beta$;
6:     $\mathcal{G}_{crossover}$ = Crossover($\mathcal{G}_i, \mathcal{S}$), s.t. $\beta$;
7:     $\mathcal{G}_{i+1} = \mathcal{G}_i + \mathcal{G}_{mutation} + \mathcal{G}_{crossover}$;
8: **end for**
9: $\boldsymbol{p}^*$, Fitness$^*$= Top1($\{\mathcal{G}_{\mathcal{N}+1}, \text{Fitness}\}$);
10: **return** $\boldsymbol{p}^*$, Fitness$^*$.

---

## 4 EXPERIMENTS

### 4.1 EXPERIMENTAL SETTINGS

**Models.** We implemented our method on LLaMA-1 (Touvron et al., 2023a), LLaMA-2 (Touvron et al., 2023b), LLaMA-3 (Meta, 2024a), LLaMA-3.1 (Meta, 2024b), and Vicuna (Chiang et al., 2023), with parameter counts ranging from 7 billion to 70 billion.

**Baselines.** We compared our method with two prior state-of-the-art pruning methods: LLM-Pruner (Ma et al., 2023) and Wanda-sp (An et al., 2024), where Wanda-sp is the structured pruning extension of the unstructured pruning method Wanda (Sun et al., 2023). All these methods applied post-training pruning to LLMs without updating the pruned model weights.

**Evaluation.** We assessed the perplexity of the pruned LLMs on the WikiText-2 (Merity et al., 2016) dataset. Additionally, we evaluated the zero-shot commonsense reasoning capability on tasks such as Winogrande (Sakaguchi et al., 2021), HellaSwag (Zellers et al., 2019), BoolQ (Clark et al., 2019), ARC-Easy, ARC-Challenge (Clark et al., 2018), OpenBookQA (Mihaylov et al., 2018), and PIQA (Bisk et al., 2020). We utilized lm-eval-harness (Gao et al., 2021) to generate commonsense question-answering results.

Table 1: Perplexity of pruned LLMs on WikiText-2 dataset at 20-50% pruning ratio.

| | | LLaMA-1 | | LLaMA-2 | | | LLaMA-3 | LLaMA-3.1 | Vicuna |
|---|---|---|---|---|---|---|---|---|---|
| Sparsity | Method | 7B | 13B | 7B | 13B | 70B | 8B | 8B | 13B |
| 0% | Dense | 5.68 | 5.09 | 5.12 | 4.57 | 3.12 | 6.05 | 6.18 | 5.94 |
| 20% | LLM-Pruner | 9.87 | 7.72 | 10.48 | 8.00 | / | 12.02 | 12.77 | 9.95 |
| | Wanda-sp | 13.46 | 7.74 | 12.01 | 7.45 | 4.27 | 12.61 | 11.35 | 9.45 |
| | **Self-Pruner** | **9.07** | **7.32** | **9.92** | **6.46** | **4.24** | **10.35** | **10.37** | **7.94** |
| 30% | LLM-Pruner | 18.42 | 11.47 | 17.90 | 11.64 | / | 20.94 | 21.17 | 13.97 |
| | Wanda-sp | 21.02 | 10.74 | 24.53 | 14.01 | 5.10 | 37.46 | 27.00 | 16.10 |
| | **Self-Pruner** | **16.64** | **9.92** | **16.25** | **10.26** | **5.00** | **16.80** | **16.79** | **11.08** |
| 40% | LLM-Pruner | 35.82 | 21.73 | 46.32 | 21.68 | / | 39.32 | 44.27 | 26.95 |
| | Wanda-sp | 40.79 | 27.53 | 38.65 | 69.86 | 6.74 | 71.97 | 58.49 | 144.68 |
| | **Self-Pruner** | **30.17** | **14.44** | **28.52** | **16.92** | **5.95** | **33.57** | **41.46** | **18.07** |
| 50% | LLM-Pruner | 111.00 | 51.14 | 253.13 | 55.81 | / | 111.90 | 121.54 | 71.18 |
| | Wanda-sp | 411.06 | 82.87 | 249.18 | 90.90 | 16.78 | 205.21 | 160.49 | 183.44 |
| | **Self-Pruner** | **59.11** | **23.24** | **53.63** | **41.95** | **9.08** | **68.40** | **64.56** | **46.47** |

* LLM-Pruner (Ma et al., 2023) employs the Taylor pruning (Molchanov et al., 2016) metric, which requires expensive gradient computations. For the LLaMA-2-70B model, we did not find relevant experimental data. Furthermore, due to the limited number of our GPU devices, we did not report its experimental results.

**Implementation Details.** All implementations were carried out on NVIDIA A100 80GB GPUs. Models with up to 30 billion parameters used a single GPU, while the 70 billion parameter model used two GPUs. The settings of all hyperparameters in evolutionary search are: population size $\mathcal{K} = 30$, number of mutation $\mathcal{M} = 10$, number of crossover $\mathcal{S} = 10$ and max number of iterations $\mathcal{N} = 20$. We employed the OpenAI GPT4-o model (OpenAI, 2024) to generate solutions for the evolutionary search.

## 4.2 LANGUAGE MODELING

We report the perplexity of pruned LLMs with pruning rates ranging from 20% to 50% on the WikiText-2 (Merity et al., 2016) dataset in Table 1. Self-Pruner significantly outperforms existing post-training pruning techniques, further narrowing the accuracy gap between post-training structured pruned LLMs and the original models, especially under high pruning rate settings. Notably, Self-Pruner is particularly beneficial for larger models, especially LLaMA-2-70B. Using the Self-Pruner method, under the 30% pruning rate setting, the model's perplexity only increased by 1.88 compared to the original model, and under the 50% pruning rate setting, the perplexity increased by only 5.96. This comprehensive superiority over existing techniques once again demonstrates that LLMs can leverage their inherent knowledge to design compressed LLMs architectures with good accuracy, proving the feasibility of LLMs automatically performing model compression tasks.

## 4.3 ZERO-SHOT TASKS

To evaluate the generalization capability of Self-Pruner, we assessed the performance of pruned LLMs in zero-shot settings across seven commonsense tasks. Table 2 shows the average performance of pruned LLMs across all seven tasks, with detailed results for each task provided in Appendix A. The experimental results demonstrate that Self-Pruner significantly outperforms existing post-training structured pruning techniques. For instance, for the LLaMA-2-7B model, Self-Pruner outperforms the Wanda-sp (Sun et al., 2023) method by 3.13% at a 30% pruning rate and outperforms the LLM-Pruner (Ma et al., 2023) method by 14.59% at a 50% pruning rate, further narrowing the performance gap with the original model. Additionally, we observed that as the model size increases, the gap in zero-shot task accuracy between the pruned models and the original model further reduces. Larger models, such as LLaMA-2-70B, there is only a 0.80% drop in accuracy at a 30% pruning rate and experience only a 3.80% drop under an extreme high pruning rate of 50%. These experimental results further validate the effectiveness of the Self-Pruner method.

Table 2: Mean zero-shot accuracy results of pruned LLMs on the Winogrande, HellaSwag, BoolQ, ARC-Easy, ARC-Challenge, OpenBookQA and PIQA datasets at 20-50% pruning ratio.

| | | LLaMA-1 | | LLaMA-2 | | | LLaMA-3 | LLaMA-3.1 | Vicuna |
|---|---|---|---|---|---|---|---|---|---|
| Sparsity | Method | 7B | 13B | 7B | 13B | 70B | 8B | 8B | 13B |
| 0% | Dense | 66.30 | 68.40 | 66.82 | 69.28 | 73.81 | 70.21 | 70.64 | 69.73 |
| 20% | LLM-Pruner | 60.14 | 64.34 | 60.19 | 64.24 | / | 56.50 | 57.45 | 65.06 |
| | Wanda-sp | 63.12 | 65.41 | 62.33 | 66.28 | 72.84 | 59.90 | 62.16 | 66.91 |
| | **Self-Pruner** | **64.80** | **66.23** | **63.44** | **67.15** | **73.11** | **61.71** | **63.61** | **68.18** |
| 30% | LLM-Pruner | 53.76 | 59.31 | 51.89 | 58.95 | / | 51.80 | 52.45 | 57.92 |
| | Wanda-sp | 59.02 | 61.71 | 57.66 | 60.94 | 72.66 | 40.91 | 47.19 | 62.35 |
| | **Self-Pruner** | **61.79** | **64.22** | **60.79** | **64.34** | **73.01** | **56.74** | **58.09** | **65.90** |
| 40% | LLM-Pruner | 46.56 | 54.02 | 46.63 | 50.59 | / | 41.93 | 42.75 | 51.74 |
| | Wanda-sp | 43.98 | 56.76 | 52.86 | 38.81 | 69.71 | 38.43 | 40.06 | 39.08 |
| | **Self-Pruner** | **58.26** | **63.15** | **57.59** | **63.22** | **71.97** | **53.30** | **53.88** | **63.35** |
| 50% | LLM-Pruner | 41.96 | 47.25 | 40.45 | 42.12 | / | 39.15 | 40.61 | 43.77 |
| | Wanda-sp | 37.88 | 43.09 | 36.76 | 39.50 | 60.63 | 37.80 | 38.43 | 39.50 |
| | **Self-Pruner** | **52.96** | **58.38** | **51.35** | **58.52** | **70.01** | **46.13** | **46.99** | **54.64** |

\* LLM-Pruner (Ma et al., 2023) employs the Taylor pruning (Molchanov et al., 2016) metric, which requires expensive gradient computations. For the LLaMA-2-70B model, we did not find relevant experimental data. Furthermore, due to the limited number of our GPU devices, we did not report its experimental results.

## 4.4 ABLATION STUDY

**Ablation of each component in Self-Pruner.** To demonstrate the effectiveness of each component in Self-Pruner, we present the final accuracy of the algorithm when each component is removed in Table 3. Specifically, we are targeting a 30% pruning rate for LLaMA-2-7B. First, instead of using LLMs to generate the initial population, we replace it with a random initial population (w/o initialization). Additionally, we remove LLMs responsible for performing mutation and crossover operations, meaning no mutation or crossover is performed during evolutionary search (w/o mutation/crossover). The results show that each component of Self-Pruner contributes positively to the final performance of the algorithm, and removing any component results in a drop in final accuracy.

Table 3: Ablation of effectiveness of each component in Self-Pruner.

| Method | Perplexity | Accuracy |
|---|---|---|
| w/o initialization | 17.43 | 59.30 |
| w/o mutation | 17.65 | 58.98 |
| w/o crossover | 17.59 | 59.15 |
| **Self-Pruner** | **16.25** | **60.79** |

**Different LLMs.** To analyze the impact of different capabilities of LLMs on the final accuracy, we compared four commonly used LLMs: GPT-3.5, GPT 4 and GPT 4o. From the experimental results in Table 4, we can observe that Self-Pruner can generate high-accuracy pruning rate using these different LLMs. However, due to the strong reasoning ability of GPT 4o, it is able to search and obtain the best pruning rate than other LLM.

Table 4: Impact of different LLMs on final accuracy.

| Model | Pruning Ratio | Perplexity | Accuracy |
|---|---|---|---|
| LLaMA-2-7B | 0% | 5.12 | 66.82 |
| GPT-3.5 | 30% | 17.44 | 58.94 |
| GPT 4 | 30% | 17.13 | 59.87 |
| **GPT 4o** | **30%** | **16.25** | **60.79** |

Table 5: Results of Self-Pruner vs. OWL.

| Model | Pruning Ratio | Perplexity | Accuracy |
|---|---|---|---|
| LLaMA-2-7B | 0% | 5.12 | 66.82 |
| OWL | 30% | 21.40 | 59.08 |
| **Self-Pruner** | **30%** | **16.25** | **60.79** |
| OWL | 50% | 81.56 | 39.06 |
| **Self-Pruner** | **50%** | **53.63** | **51.35** |

## 4.5 ANALYSIS

**Self-Pruner vs. OWL.** We further compared another method for determining the layer-wise pruning rates of LLMs: OWL (Yin et al., 2023). We applied OWL to determine the layer-wise pruning rates of pruned LLMs and use the Wanda-sp (Sun et al., 2023) metric to decide which components within the layers should be pruned like Self-Pruner. Two important hyperparameters in OWL follow the settings in the paper, which are $\lambda = 0.08, M = 5$. The results in Table 5 indicate that Self-Pruner outperforms OWL across different pruning rates. This demonstrates that the manual determination of layer importance metrics is suboptimal, whereas Self-Pruner, by utilizing LLMs to assist in evolutionary search, minimizes the impact of manual intervention on the final accuracy and automatically finds the good layer-wise pruning rate.

**LoRA Fine-tuning.** Due to the severe accuracy degradation caused by structured pruning at high pruning rates, we further demonstrate the potential of fine-tuning to mitigate performance loss in pruned LLMs. Specifically, we apply the LoRA (Hu et al., 2021) (r = 8) method for to fine-tune the pruned LLaMA-2-7B. We randomly sampled a 10K subset from the Alpaca-GPT4 (Peng et al., 2023) dataset as our fine-tuning dataset. The experimental results in Table 6 show that LoRA fine-tuning can recover the performance of pruned LLMs, further narrowing the performance gap with the original model.

Table 6: Experimental results of restoring the accuracy of pruned LLaMA-2-7B using LoRA fine-tuning.

| Method | Pruning Rate | Perplexity | Accuracy |
|---|---|---|---|
| Self-Pruner | 30% | 16.25 | 60.79 |
| **w. LoRA** | **30**% | **11.87** | **63.01** |
| Self-Pruner | 50% | 53.63 | 51.35 |
| **w. LoRA** | **50**% | **25.32** | **59.78** |

**Inference Speedup and GPU Memory Usage.** We report the parameter count, GPU memory usage, and inference speedup of structural pruning LLMs in Table 7. The results were obtained using the vLLM inference engine (Kwon et al., 2023) on NVIDIA A100 80GB GPUs. Thanks to the hardware-friendly nature of structured pruning, it effectively reduces the number of model parameters, lowers GPU memory usage, and achieves up to $1.82\times$ inference speedup.

Table 7: Inference speedup and GPU memory usage statistics of structured pruning LLMs.

| Model | Pruning Ratio | Params (B) | Memory (GB) | Tokens/s | Speedup |
|---|---|---|---|---|---|
| | 0% | 6.74 | 12.55 | 83.77 | **1.00**× |
| | 20% | 5.47 | 10.20 | 115.39 | **1.23**× |
| LLaMA-2-7B | 30% | 4.77 | 8.88 | 123.66 | **1.48**× |
| | 40% | 4.13 | 7.80 | 137.11 | **1.64**× |
| | 50% | 3.50 | 6.55 | 152.24 | **1.82**× |
| | 0% | 68.98 | 128.48 | 18.44 | **1.00**× |
| | 20% | 55.53 | 103.43 | 22.13 | **1.20**× |
| LLaMA-2-70B | 30% | 48.56 | 90.40 | 25.63 | **1.39**× |
| | 40% | 41.59 | 77.50 | 28.58 | **1.55**× |
| | 50% | 34.75 | 64.64 | 31.31 | **1.70**× |

## 5 LIMITATIONS AND FUTURE WORK

The limitations of existing methods mainly include two points: 1) The complete automation of LLMs compression has not yet been achieved. In this paper, we take the first step towards LLMs self-compression by using LLMs to generate solutions for evolutionary algorithm to search for layer-wise pruning rate configurations. However, layer-wise pruning rates alone are not sufficient to achieve fully compressed LLMs, indicating that our study is still some distance away from enabling LLMs to perform compression tasks on their own. In the future, we aim to further automate the LLMs pruning process. 2) Post-training structured pruning of LLMs still falls short of the original model's accuracy. Post-training pruning significantly reduces LLM accuracy. Although this study has narrowed this gap, the accuracy of pruned LLMs remains low at high pruning rates. In the future, we will explore efficient fine-tuning methods for LLMs to restore the accuracy of pruned models.

## 6 CONCLUSION

In this paper, we propose Self-Pruner, an novel framework that automatically finds layer-wise pruning rates for LLMs using an evolutionary search driven by LLMs. Self-Pruner enables LLMs to execute the evolutionary search process themselves, capitalizing on their prior knowledge of model redundancy to generate, evaluate, and optimize pruning rates. By integrating LLMs into the search process, Self-Pruner accelerates the convergence of the evolutionary search, reducing the need for extensive human intervention in evolutionary algorithm design. Extensive experimental results demonstrate that Self-Pruner significantly enhances the accuracy of post-training structural pruning for LLMs.

## REPRODUCIBILITY STATEMENTS

The experimental setup is detailed in Section 4.1. We have included the code to reproduce our results in the supplementary materials, and we plan to release the code publicly.

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

# A    DETAILED ZERO-SHOT TASK RESULTS

To further enhance our comprehensive understanding of the performance of pruned LLMs, we present experimental data of pruned LLMs across all seven commonsense tasks in this section. These datasets include Winogrande (Sakaguchi et al., 2021), HellaSwag (Zellers et al., 2019), BoolQ (Clark et al., 2019), ARC-Easy, ARC-Challenge (Clark et al., 2018), OpenBookQA (Mihaylov et al., 2018), and PIQA (Bisk et al., 2020). We show the experimental data for pruning rates ranging from 20% to 50% in Tables 8, 9, 10, and 11, respectively. From the experimental data, we observe that Self-Pruner consistently enhances the capabilities of pruned LLMs, achieving higher accuracy compared to existing techniques.

Table 8: Zero-shot performance of pruned LLMs with 20% pruning ratio.

| Model | Method | HellaSwag | Winogrande | BoolQ | OBQA | PIQA | ARC-e | ARC-c | Mean |
|---|---|---|---|---|---|---|---|---|---|
| LLaMA-1-7B | Dense | 76.19 | 69.85 | 75.11 | 44.40 | 78.62 | 75.25 | 44.71 | 66.30 |
| | LLM-Pruner | 69.85 | 62.51 | 64.56 | 41.60 | 75.57 | 68.01 | 38.91 | 60.14 |
| | Wanda-sp | 72.70 | 66.93 | 71.47 | 40.20 | 76.33 | 71.80 | 42.41 | 63.12 |
| | **Self-Pruner** | **73.67** | **68.58** | **75.57** | **43.60** | **77.31** | **72.39** | **42.49** | **64.80** |
| LLaMA-1-13B | Dense | 79.06 | 72.69 | 77.86 | 44.80 | 79.16 | 77.36 | 47.87 | 68.40 |
| | LLM-Pruner | 75.72 | 68.27 | 71.13 | 45.00 | 77.69 | 71.13 | 41.47 | 64.34 |
| | Wanda-sp | 77.28 | 70.56 | 71.47 | 43.40 | 78.18 | 72.22 | 44.80 | 65.41 |
| | **Self-Pruner** | **77.93** | **72.06** | **72.14** | **45.00** | **78.56** | **72.85** | **45.05** | **66.23** |
| LLaMA-2-7B | Dense | 75.97 | 69.06 | 77.74 | 44.20 | 78.07 | 76.35 | 46.33 | 66.82 |
| | LLM-Pruner | 68.72 | 63.54 | 65.14 | **39.80** | 75.90 | 68.81 | 39.42 | 60.19 |
| | Wanda-sp | 72.86 | 65.98 | 67.61 | 38.80 | 76.61 | 72.18 | 42.24 | 62.33 |
| | **Self-Pruner** | **73.51** | **68.98** | **69.63** | 39.20 | **76.93** | **72.19** | **43.60** | **63.44** |
| LLaMA-2-13B | Dense | 79.38 | 72.22 | 80.55 | 45.20 | 79.11 | 79.42 | 49.06 | 69.28 |
| | LLM-Pruner | 74.65 | 65.74 | 67.77 | **43.80** | 78.29 | 73.06 | 46.33 | 64.24 |
| | Wanda-sp | 77.60 | 69.69 | 73.58 | 43.40 | 78.40 | 75.84 | 45.48 | 66.28 |
| | **Self-Pruner** | **78.42** | **70.40** | **76.76** | 42.40 | **78.56** | **76.85** | **46.67** | **67.15** |
| LLaMA-2-70B | Dense | 83.81 | 77.90 | 83.73 | 48.80 | 82.21 | 82.74 | 57.51 | 73.81 |
| | Wanda-sp | 83.15 | 77.51 | 83.15 | 47.40 | **82.05** | 81.00 | 55.63 | 72.84 |
| | **Self-Pruner** | **83.85** | **77.59** | **83.70** | **47.60** | 81.92 | **81.45** | **55.66** | **73.11** |
| LLaMA-3-8B | Dense | 79.16 | 72.77 | 81.35 | 45.00 | 79.71 | 80.09 | 53.41 | 70.21 |
| | LLM-Pruner | 62.15 | 58.41 | 59.54 | 37.80 | 75.08 | 66.62 | 35.92 | 56.50 |
| | Wanda-sp | 65.45 | 66.93 | **64.46** | 39.00 | 76.28 | 68.60 | 38.57 | 59.90 |
| | **Self-Pruner** | **71.24** | **68.27** | 61.10 | **41.60** | **78.13** | **71.09** | **40.53** | **61.71** |
| LLaMA-3.1-8B | Dense | 79.01 | 73.32 | 82.05 | 45.00 | 79.98 | 81.57 | 53.58 | 70.64 |
| | LLM-Pruner | 61.02 | 59.27 | 62.45 | 38.40 | 74.81 | 68.90 | 37.29 | 57.45 |
| | Wanda-sp | 68.91 | 67.64 | 66.91 | 41.00 | 77.31 | 71.72 | 41.64 | 62.16 |
| | **Self-Pruner** | **71.04** | **69.22** | **67.46** | **43.00** | **78.13** | **73.06** | **43.34** | **63.61** |
| Vicuna-13B | Dense | 77.49 | 71.59 | 85.26 | 45.40 | 79.00 | 78.66 | 50.68 | 69.73 |
| | LLM-Pruner | 72.54 | 66.30 | 71.99 | **45.40** | 77.75 | 75.21 | 46.25 | 65.06 |
| | Wanda-sp | 75.10 | 68.75 | 79.51 | 44.00 | 77.37 | 76.18 | 47.44 | 66.91 |
| | **Self-Pruner** | **76.80** | **69.69** | **82.87** | 44.00 | **78.40** | **77.69** | **47.78** | **68.18** |

Table 9: Zero-shot performance of pruned LLMs with 30% pruning ratio.

| Model | Method | HellaSwag | Winogrande | BoolQ | OBQA | PIQA | ARC-e | ARC-c | Mean |
|---|---|---|---|---|---|---|---|---|---|
| LLaMA-1-7B | Dense | 76.19 | 69.85 | 75.11 | 44.40 | 78.62 | 75.25 | 44.71 | 66.30 |
| | LLM-Pruner | 61.27 | 58.64 | 53.33 | 38.60 | 72.03 | 58.04 | 34.39 | 53.76 |
| | Wanda-sp | 67.91 | 62.12 | 66.97 | 38.60 | 74.16 | 65.74 | 37.63 | 59.02 |
| | **Self-Pruner** | **70.96** | **66.30** | **72.51** | **39.80** | **75.57** | **67.17** | **40.19** | **61.79** |
| LLaMA-1-13B | Dense | 79.06 | 72.69 | 77.86 | 44.80 | 79.16 | 77.36 | 47.87 | 68.40 |
| | LLM-Pruner | 70.70 | 63.22 | 62.39 | 42.20 | 75.73 | 62.12 | 38.82 | 59.31 |
| | Wanda-sp | 74.17 | 66.06 | 65.20 | 42.00 | 76.82 | 67.72 | 40.02 | 61.71 |
| | **Self-Pruner** | **76.79** | **71.51** | **67.25** | **43.00** | **77.64** | **69.95** | **43.43** | **64.22** |
| LLaMA-2-7B | Dense | 75.97 | 69.06 | 77.74 | 44.20 | 78.07 | 76.35 | 46.33 | 66.82 |
| | LLM-Pruner | 57.77 | 55.49 | 50.12 | 36.80 | 71.87 | 58.84 | 32.34 | 51.89 |
| | Wanda-sp | 65.94 | 58.33 | 63.12 | 38.00 | 74.76 | 65.99 | 37.46 | 57.66 |
| | **Self-Pruner** | **70.63** | **67.40** | **64.71** | **40.20** | **75.30** | **67.85** | **39.42** | **60.79** |
| LLaMA-2-13B | Dense | 79.38 | 72.22 | 80.55 | 45.20 | 79.11 | 79.42 | 49.06 | 69.28 |
| | LLM-Pruner | 68.31 | 60.38 | 57.09 | 43.40 | 76.12 | 66.66 | 40.70 | 58.95 |
| | Wanda-sp | 71.59 | 63.54 | 66.94 | 39.40 | 76.33 | 68.64 | 40.10 | 60.94 |
| | **Self-Pruner** | **75.81** | **70.09** | **69.94** | **41.00** | **77.53** | **72.73** | **43.26** | **64.34** |
| LLaMA-2-70B | Dense | 83.81 | 77.90 | 83.73 | 48.80 | 82.21 | 82.74 | 57.51 | 73.81 |
| | Wanda-sp | 81.45 | **78.27** | 84.20 | 47.40 | 81.66 | 80.95 | 54.66 | 72.66 |
| | **Self-Pruner** | **82.75** | 78.06 | **84.22** | **48.20** | **81.73** | **81.35** | **54.76** | **73.01** |
| LLaMA-3-8B | Dense | 79.16 | 72.77 | 81.35 | 45.00 | 79.71 | 80.09 | 53.41 | 70.21 |
| | LLM-Pruner | 54.16 | 56.12 | **52.87** | 36.40 | 71.93 | 58.63 | 32.51 | 51.80 |
| | Wanda-sp | 36.96 | 51.22 | 40.89 | 26.80 | 62.46 | 43.01 | 25.00 | 40.91 |
| | **Self-Pruner** | **65.21** | **64.64** | 49.94 | **39.00** | **76.12** | **65.87** | **36.43** | **56.74** |
| LLaMA-3.1-8B | Dense | 79.01 | 73.32 | 82.05 | 45.00 | 79.98 | 81.57 | 53.58 | 70.64 |
| | LLM-Pruner | 53.22 | 57.93 | **54.89** | 33.20 | 72.14 | 61.95 | 33.87 | 52.45 |
| | Wanda-sp | 48.52 | 53.75 | 52.11 | 30.20 | 66.38 | 51.85 | 27.56 | 47.19 |
| | **Self-Pruner** | **66.07** | **65.59** | 53.61 | **39.60** | **76.99** | **67.26** | **37.54** | **58.09** |
| Vicuna-13B | Dense | 77.49 | 71.59 | 85.26 | 45.40 | 79.00 | 78.66 | 50.68 | 69.73 |
| | LLM-Pruner | 65.62 | 59.27 | 53.15 | 42.20 | 75.57 | 68.43 | 41.21 | 57.92 |
| | Wanda-sp | 69.19 | 62.67 | 72.94 | 39.40 | 75.95 | 71.76 | 44.52 | 62.35 |
| | **Self-Pruner** | **74.35** | **68.98** | **81.01** | **42.30** | **76.39** | **73.78** | **44.54** | **65.90** |

Table 10: Zero-shot performance of pruned LLMs with 40% pruning ratio.

| Model | Method | HellaSwag | Winogrande | BoolQ | OBQA | PIQA | ARC-e | ARC-c | Mean |
|---|---|---|---|---|---|---|---|---|---|
| LLaMA-1-7B | Dense | 76.19 | 69.85 | 75.11 | 44.40 | 78.62 | 75.25 | 44.71 | 66.30 |
| | LLM-Pruner | 49.34 | 54.62 | 45.08 | 33.20 | 67.36 | 46.55 | 29.78 | 46.56 |
| | Wanda-sp | 39.45 | 53.51 | 59.39 | 28.60 | 62.40 | 38.47 | 26.02 | 43.98 |
| | **Self-Pruner** | **66.49** | **64.09** | **68.44** | **37.20** | **72.52** | **61.20** | **37.88** | **58.26** |
| LLaMA-1-13B | Dense | 79.06 | 72.69 | 77.86 | 44.80 | 79.16 | 77.36 | 47.87 | 68.40 |
| | LLM-Pruner | 61.60 | 57.77 | 60.92 | 37.60 | 72.85 | 53.37 | 34.04 | 54.02 |
| | Wanda-sp | 67.24 | 59.75 | 62.75 | 34.20 | 73.83 | 62.12 | 37.46 | 56.76 |
| | **Self-Pruner** | **74.29** | **70.01** | **70.34** | **41.80** | **75.46** | **67.17** | **43.00** | **63.15** |
| LLaMA-2-7B | Dense | 75.97 | 69.06 | 77.74 | 44.20 | 78.07 | 76.35 | 46.33 | 66.82 |
| | LLM-Pruner | 43.48 | 52.80 | 59.82 | 32.20 | 65.56 | 42.59 | 29.95 | 46.63 |
| | Wanda-sp | 57.31 | 54.70 | 61.56 | 35.40 | 70.62 | 56.99 | 33.45 | 52.86 |
| | **Self-Pruner** | **65.13** | **64.25** | **62.72** | **39.00** | **73.72** | **60.52** | **37.80** | **57.59** |
| LLaMA-2-13B | Dense | 79.38 | 72.22 | 80.55 | 45.20 | 79.11 | 79.42 | 49.06 | 69.28 |
| | LLM-Pruner | 57.59 | 55.64 | 42.45 | 39.40 | 71.16 | 54.76 | 33.11 | 50.59 |
| | Wanda-sp | 28.50 | 50.04 | 62.14 | 24.00 | 54.90 | 28.45 | 23.63 | 38.81 |
| | **Self-Pruner** | **73.39** | **68.59** | **75.17** | **40.20** | **75.14** | **66.96** | **43.09** | **63.22** |
| LLaMA-2-70B | Dense | 83.81 | 77.90 | 83.73 | 48.80 | 82.21 | 82.74 | 57.51 | 73.81 |
| | Wanda-sp | 80.60 | 74.03 | 80.45 | 45.20 | 79.98 | 77.30 | 50.43 | 69.71 |
| | **Self-Pruner** | **81.00** | **78.53** | **84.80** | **48.20** | **80.85** | **78.90** | **51.53** | **71.97** |
| LLaMA-3-8B | Dense | 79.16 | 72.77 | 81.35 | 45.00 | 79.71 | 80.09 | 53.41 | 70.21 |
| | LLM-Pruner | 37.76 | 52.17 | 42.45 | 28.80 | 64.36 | 42.09 | 25.85 | 41.93 |
| | Wanda-sp | 30.16 | 49.88 | **52.66** | 25.00 | 56.53 | 33.42 | 21.33 | 38.43 |
| | **Self-Pruner** | **59.24** | **62.83** | 51.16 | **37.80** | **73.45** | **56.57** | **32.08** | **53.30** |
| LLaMA-3.1-8B | Dense | 79.01 | 73.32 | 82.05 | 45.00 | 79.98 | 81.57 | 53.58 | 70.64 |
| | LLM-Pruner | 38.12 | 52.96 | 43.09 | 29.80 | 65.56 | 45.24 | 24.49 | 42.75 |
| | Wanda-sp | 31.86 | 52.57 | **57.09** | 25.20 | 57.78 | 33.84 | 22.10 | 40.06 |
| | **Self-Pruner** | **60.32** | **63.22** | 46.73 | **39.40** | **73.78** | **59.09** | **34.64** | **53.88** |
| Vicuna-13B | Dense | 77.49 | 71.59 | 85.26 | 45.40 | 79.00 | 78.66 | 50.68 | 69.73 |
| | LLM-Pruner | 56.99 | 54.78 | 49.54 | 39.00 | 72.09 | 55.18 | 34.64 | 51.74 |
| | Wanda-sp | 28.49 | 51.14 | 61.93 | 26.40 | 54.13 | 29.17 | 22.27 | 39.08 |
| | **Self-Pruner** | **71.75** | **66.85** | **80.40** | **39.10** | **74.37** | **69.49** | **41.55** | **63.35** |

Table 11: Zero-shot performance of pruned LLMs with 50% pruning ratio.

| Model | Method | HellaSwag | Winogrande | BoolQ | OBQA | PIQA | ARC-e | ARC-c | Mean |
|-------|--------|-----------|------------|-------|------|------|-------|-------|------|
| LLaMA-1-7B | Dense | 76.19 | 69.85 | 75.11 | 44.40 | 78.62 | 75.25 | 44.71 | 66.30 |
| | LLM-Pruner | 36.64 | 52.96 | 51.07 | 30.20 | 61.81 | 35.19 | 25.85 | 41.96 |
| | Wanda-sp | 29.39 | 50.83 | 54.13 | 23.40 | 54.79 | 29.84 | 22.78 | 37.88 |
| | Self-Pruner | 59.35 | 59.67 | 63.06 | 33.20 | 69.75 | 52.95 | 32.76 | 52.96 |
| LLaMA-1-13B | Dense | 79.06 | 72.69 | 77.86 | 44.80 | 79.16 | 77.36 | 47.87 | 68.40 |
| | LLM-Pruner | 47.16 | 53.43 | 61.01 | 35.20 | 67.14 | 37.79 | 29.01 | 47.25 |
| | Wanda-sp | 38.86 | 51.54 | 57.92 | 28.00 | 61.43 | 38.47 | 25.43 | 43.09 |
| | Self-Pruner | 69.68 | 66.85 | 65.66 | 37.20 | 72.25 | 58.96 | 38.05 | 58.38 |
| LLaMA-2-7B | Dense | 75.97 | 69.06 | 77.74 | 44.20 | 78.07 | 76.35 | 46.33 | 66.82 |
| | LLM-Pruner | 31.41 | 50.43 | 57.92 | 28.00 | 57.73 | 31.65 | 26.02 | 40.45 |
| | Wanda-sp | 27.88 | 48.93 | 48.53 | 25.40 | 54.73 | 29.50 | 22.35 | 36.76 |
| | Self-Pruner | 55.64 | 53.12 | 60.73 | 36.40 | 69.26 | 51.68 | 32.59 | 51.35 |
| LLaMA-2-13B | Dense | 79.38 | 72.22 | 80.55 | 45.20 | 79.11 | 79.42 | 49.06 | 69.28 |
| | LLM-Pruner | 43.45 | 52.88 | 38.26 | 33.20 | 64.31 | 35.90 | 26.88 | 42.12 |
| | Wanda-sp | 30.23 | 50.20 | 62.02 | 26.00 | 54.57 | 27.78 | 25.68 | 39.50 |
| | Self-Pruner | 66.26 | 66.38 | 64.77 | 39.00 | 71.55 | 59.97 | 41.72 | 58.52 |
| LLaMA-2-70B | Dense | 83.81 | 77.90 | 83.73 | 48.80 | 82.21 | 82.74 | 57.51 | 73.81 |
| | Wanda-sp | 68.30 | 58.96 | 67.85 | 40.20 | 76.39 | 71.25 | 41.47 | 60.63 |
| | Self-Pruner | 79.15 | 78.85 | 82.85 | 45.00 | 79.11 | 75.65 | 49.48 | 70.01 |
| LLaMA-3-8B | Dense | 79.16 | 72.77 | 81.35 | 45.00 | 79.71 | 80.09 | 53.41 | 70.21 |
| | LLM-Pruner | 33.22 | 49.81 | 41.53 | 29.00 | 60.45 | 36.32 | 23.72 | 39.15 |
| | Wanda-sp | 29.03 | 50.91 | 50.09 | 25.00 | 55.88 | 31.57 | 22.10 | 37.80 |
| | Self-Pruner | 47.33 | 55.88 | 41.50 | 32.80 | 67.79 | 48.11 | 29.52 | 46.13 |
| LLaMA-3.1-8B | Dense | 79.01 | 73.32 | 82.05 | 45.00 | 79.98 | 81.57 | 53.58 | 70.64 |
| | LLM-Pruner | 32.45 | 52.64 | 49.60 | 28.00 | 60.34 | 38.05 | 23.21 | 40.61 |
| | Wanda-sp | 28.29 | 49.33 | 57.68 | 25.00 | 56.20 | 30.47 | 22.01 | 38.43 |
| | Self-Pruner | 48.95 | 56.75 | 45.57 | 30.80 | 67.08 | 49.83 | 29.95 | 46.99 |
| Vicuna-13B | Dense | 77.49 | 71.59 | 85.26 | 45.40 | 79.00 | 78.66 | 50.68 | 69.73 |
| | LLM-Pruner | 44.59 | 52.01 | 43.15 | 31.80 | 65.29 | 40.74 | 28.84 | 43.77 |
| | Wanda-sp | 29.92 | 51.14 | 62.08 | 24.80 | 53.97 | 28.45 | 26.11 | 39.50 |
| | Self-Pruner | 59.55 | 59.67 | 63.46 | 33.60 | 70.35 | 58.96 | 36.86 | 54.64 |

