# OpenReview forum: "Towards Efficient Automatic Self-Pruning of Large Language Models"
_ICLR.cc/2025/Conference — Submitted to ICLR 2025_

### Official Review · Reviewer_XhYR · 2024-10-28

**Soundness:** 2
**Presentation:** 2
**Contribution:** 2
**Rating:** 5
**Confidence:** 5

**Summary:**

This paper utilized the power of LLM to search for the best sparsity ratio using evolution search. GPT-4 is utilized for crossover and mutation; perplexity is employed as the fitness.

**Strengths:**

- This paper is easy to follow;
- This paper employs LLM to search for the sparsity ratio, which sounds interesting.
- Under 30% pruning ratio, self-pruner can prune 70b to 49b, achieving 1.39 speedup.

**Weaknesses:**

- Firstly, the method name - self-pruner - is really confusing. It is difficult to understand the meaning of self-pruning. From my understanding, each pruning methods, including unstructured, structured and semi-structured pruning should be identified to self-pruning.
- From table 1 and table 2, I notice that there is a huge ppl increase at high pruning ratio. In most cases, when the ppl is higher than 10, it can not speak normally. So can you provide me some examples of its answers to several common questions?
- About the experiment settings: I think there is space for improving your experiments designs. (1) You missed several important structured pruning method like SliceGPT, FLAP, SLEB, Shortened LLaMA etc. Please add experiments for elucidate it. (2) OWL should be the most important baseline, you should compare it in each table not just in ablation study.
- About the evaluation cost, I think the key bottleneck of this approach is the searching cost. In this paper, I haven't find the search cost of each size of Llama models. Based on my experience, I have tested the inference time of evaluating the ppl of LLaMA 70B. It takes me a lot of time to load and compute it. When the search space is exploding, it is impractical to use this kind of evaluation.

**Questions:**

- Can you show me the evaluation cost on LLama-70B?
- Under 50% sparsity ratio, I noticed that the perplexity is increasing significantly while the accuracy is not influenced. How to explain it?
- The proposed techniques can be utilized on unstructured pruning. Have you ever tested it?
- After reading you paper, I still do not understand how you prune llm in a structured way. Then, I checked your supplementary material and find that in lib/prune.py, you utilized wanda_sp to prune models. You set mask in column and make the column with low importance zero. I didn't find other operation that can make the pruned matrix dense again. I wonder how did you achieve the speedup?

---

### Official Review · Reviewer_UhEg · 2024-11-03

**Soundness:** 3
**Presentation:** 3
**Contribution:** 3
**Rating:** 3
**Confidence:** 3

**Summary:**

This paper proposes Self-Pruner to automatically find layer-wise pruning rates for LLMs via evolutionary search.
Particularly, Self-Pruner leverages LLMs to execute the entire evolutionary search process, including population generation, selection, crossover, and mutation, enabling the selfpruning of LLMs.
Experimental results show that Self-Pruner outperforms existing post-training pruning methods.

**Strengths:**

1. Self-Pruner reduces human-effort through using LLMs to perform mutation and crossover operations.

**Weaknesses:**

1. Since evolutionary algorithms have been previously applied to CNNs and transformers for pruning, it would be beneficial for the authors to elaborate on the specific novelty and technical contributions of their approach in the context of pruning LLMs. How does this approach differ from existing methods? Additionally, it would be useful to highlight any unique aspects of using LLMs to execute the mutation/crossover in evolutionary search process and how this might be innovative in the context of LLM pruning.
2. Evaluating pruned LLMs is time-consuming, so it would be helpful for the authors to provide a comparison of the number of evaluations required by Self-Pruner versus other pruning methods. Additionally, an analysis of the trade-off between evaluation time and pruning performance would be valuable. Are there any techniques employed to mitigate the evaluation time issue?
3. There is no runtime comparison with baselines, including LLM-Pruner and Wanda-SP. It would be better to include total pruning time, time per iteration, or scalability analysis with model size. Previous EA-based pruning methods are time-consuming, so providing the runtime metrics is necessary to demonstrate the superiority of Self-Pruner.

**Questions:**

1. In the experiment setup, the maximum number of iterations is 20. Can an LLM find the optimal pruning ratio with such a limited number of evolutions?
2. Given that Self-Pruner evaluates pruned LLM performance at each iteration, what is the total runtime of this algorithm?
3. How does the performance of Self-Pruner compare with the latest FLAP[1]?

Reference:
1. An, Yongqi, et al. "Fluctuation-based adaptive structured pruning for large language models." Proceedings of the AAAI Conference on Artificial Intelligence. Vol. 38. No. 10. 2024.

---

### Official Review · Reviewer_KVnL · 2024-11-03

**Soundness:** 2
**Presentation:** 3
**Contribution:** 2
**Rating:** 3
**Confidence:** 4

**Summary:**

This paper presents Self-Pruner, an automated framework for post-training structured pruning of large language models (LLMs) without requiring retraining. It leverages evolutionary algorithms, with the LLMs themselves generating, refining, and evaluating layer-wise pruning rates to optimize performance and minimize model size.

**Strengths:**

- This paper leverages LLMs to autonomously guide the pruning process using evolutionary algorithms.
- The experimental results that Self-Pruner achieves considerable inference speedups (up to 1.7×) with minimal accuracy loss, outperforming state-of-the-art pruning methods like LLM-Pruner and Wanda-sp.
- This paper is easy to read and understand.

**Weaknesses:**

1. Unfair Comparison with OWL: The paper does not provide a fair comparison with OWL, a foundational work in sparsity distribution for large models. OWL should be included as a baseline in Table 1 and Table 2, not merely as part of a minor ablation study, given its significance in structured pruning research.

2. Lack of Comparison with Key Related Work: The paper omits comparisons with several recent structured pruning studies, all of which were published before ICLR's submission deadline. These works include:

    [1]Shortened LLaMA: A Simple Depth Pruning for Large Language Models
    [2]SliceGPT: Compress Large Language Models by Deleting Rows and Columns
    [3]ShortGPT: Layers in Large Language Models are More Redundant Than You Expect
    [4]LaCo: Large Language Model Pruning via Layer Collapse
    [5]FLAP: Fluctuation-based adaptive structured pruning for large language models
    [6]Bonsai: Everybody Prune Now: Structured Pruning of LLMs with only Forward Passes
    [7]SLEB: Streamlining LLMs through Redundancy Verification and Elimination of Transformer Blocks
    [8]Sheared LLaMA: Accelerating Language Model Pre-training via Structured Pruning

    These omissions reduce the contextual relevance of the paper’s contributions.

3. Algorithm Stability: The framework leverages GPT-4 for evolutionary steps like crossover and mutation, but the inherent variability of large language models (e.g., parameters like temperature and top-k sampling) can affect results. The paper lacks an analysis of stability—how sensitive results are to these parameters—and omits details on specific settings (e.g., temperature, top-k) and token consumption during search.

4. Lack of Comparison with Traditional Evolutionary Algorithms: While using LLMs for mutation and crossover introduces novelty, the paper does not compare the proposed method with traditional evolutionary algorithms like genetic programming. Without such a comparison, it is unclear if the benefits justify the high computational cost of involving LLMs over simpler code-level algorithms.

5. Fitness Evaluation Cost: Calculating the fitness (e.g., perplexity) for a large model like LLaMA-70B is prohibitively expensive, especially with a large search space—discrete sparsity levels ranging from 0.1 to 0.9 for N layers result in a search space of N^9 configurations. When N=80 for LLama-70B, the search space size can be as large as 134,217,728,000,000,000. Evaluating each configuration's perplexity adds a significant computational burden, and the paper does not disclose the total time spent on these evaluations, which could be a critical bottleneck for practical use.

**Questions:**

Please address the weaknesses in the previous section.

---

### Official Review · Reviewer_MPBu · 2024-11-09

**Soundness:** 2
**Presentation:** 3
**Contribution:** 2
**Rating:** 3
**Confidence:** 3

**Summary:**

The paper proposes Self-Pruner, a framework for post-training pruning of Large Language Models (LLMs) that uses an LLM to generate the first step in an evolutionary process for determining layer-wise pruning rates. The core idea is to leverage an LLM’s intrinsic redundancy knowledge to initialize pruning configurations and facilitate evolutionary steps like crossover and mutation, leading to a smaller, faster LLM without retraining. The authors report reductions in LLM sizes with minimal accuracy loss, citing experiments on the LLaMA and Vicuna model families.

**Strengths:**

- The method appears computationally efficient and manages to avoid the need for retraining, a positive step in terms of practical utility.
- The empirical results demonstrate measurable performance improvements in inference speed and memory reduction, which are beneficial for deployment.
- Use of LLM in Evolutionary Algorithms: The idea of involving LLMs in optimization processes is novel in application, although its implementation here lacks rigor.

**Weaknesses:**

- The use of an LLM for population initialization and mutation/crossover is not sufficiently novel, as it represents a straightforward adaptation rather than a novel technique. There is no detailed exploration of why an LLM is more suited to this than simpler initialization methods.
- The evolutionary process is standard, with no significant adaptations tailored for LLMs. There’s also minimal effort to explain why this process would be more effective or yield better performance gains than other initialization methods.
- The argument that LLMs possess intrinsic redundancy knowledge is speculative and lacks empirical validation, making the approach feel more ad hoc than rigorously justified.
- The paper would benefit from a comparison with more baseline methods (e.g., using random or heuristic-based initializations without LLM assistance), which would clarify if the LLM-based method actually adds value.

**Questions:**

1. Given the modest role of the LLM in this framework, could simpler heuristic initialization methods achieve  comparable results?
2. How was the hypothesis tested that LLMs have intrinsic knowledge of their own redundancy? If not tested, what led the authors to make this assumption?
3. Could the authors explain why traditional evolutionary algorithms (without an LLM in the loop) wouldn’t perform equally well, if not better, given the limited novelty here?
4. Was there any analysis done on why using an LLM for evolutionary search convergence was better than random or heuristic-based methods?
5. Why was GPT-4 specifically chosen, and would other LLMs, even significantly smaller models, perform equally well in this initialization role?

---

### Author Response · Authors · 2024-12-03
**Global Respone**

We sincerely appreciate the reviewers' thoughtful comments and valuable suggestions. We will improve our work and resubmit.

---

### Meta-Review · Area_Chair_EgAr · 2024-12-21

**Metareview:**

**Summary:** The paper introduces Self-Pruner, an automatic post-training structured pruning framework for LLMs, leveraging evolutionary algorithms guided by LLMs. By using LLMs to generate, mutate, and evaluate pruning configurations, the method enables self-pruning with minimal human intervention.

**Strength:**

1. Automating pruning through LLM-guided evolutionary search reduces manual effort, making the method deployment-friendly.

**Weakness:**

1. Limited novelty: The evolutionary algorithm's application to pruning sounds incremental, and the hypothesis of LLMs' intrinsic redundancy knowledge lacks empirical support.

2. High evaluation cost: High computational costs for fitness evaluation, with insufficient runtime analysis, raise concerns about practical scalability.

3. Insufficient baseline comparisons: Limited comparisons with key related works (e.g., SliceGPT, FLAP, OWL) weaken claims of superiority.

**Reasons for the decision:**

The lack of comprehensive baseline comparisons and insufficient analysis of computational efficiency undermine the impact of the results. Furthermore, the novelty is incremental, and the empirical validation of the core assumptions is limited. Therefore, I recommend rejecting this paper.

**Additional Comments On Reviewer Discussion:**

The authors did not provide a rebuttal and decided to resubmit to another venue based on their comments.

---

### Decision · Program_Chairs · 2025-01-22

Reject